# Surgical Approaches to Pancoast Tumors

**DOI:** 10.3390/jpm13071168

**Published:** 2023-07-21

**Authors:** Francesco Petrella, Monica Casiraghi, Luca Bertolaccini, Lorenzo Spaggiari

**Affiliations:** 1Division of Thoracic Surgery, IRCCS European Institute of Oncology, 20141 Milan, Italy; monica.casiraghi@unimi.it (M.C.); luca.bertolaccini@ieo.it (L.B.);; 2Department of Oncology and Hemato-Oncology, University of Milan, 20122 Milan, Italy

**Keywords:** non-small-cell lung cancer, transmanubrial approach, transclavicular approach, anterior approach, chest wall, Paulson approach

## Abstract

Pancoast tumors, also defined as superior sulcus tumors, still represent a complex clinical condition requiring high technical surgical skills within more articulated multimodality treatment. The morbidity and mortality rates after Pancoast tumor treatments range from 10 to 55% and 0 to 7%, respectively, and the 5-year survival rate has significantly improved in recent years thanks to the advancement of treatments. Although a multimodality approach combining chemotherapy, radiotherapy, and surgery allows for radical resection and effective local control in the vast majority of patients, many patients cannot receive surgical resection or complete the whole programmed therapeutic regimen. Systemic relapse, particularly cerebral recurrence, still poses a significant issue in this cohort of patients. Surgical resection still plays a pivotal role within the multimodality approach. Here, we focus on surgical approaches to both anterior and posterior Pancoast tumors: the anterior transclavicular approach (Dartevelle); the anterior transmanubrial approach (Grunenwald–Spaggiari); the anterior trap-door approach (Masaoka, Nomori); the posterior approach (Shaw–Paulson); the hemiclamshell approach; and hybrid approaches. Global clinical condition, tumor histology, and long-term perspectives should always be taken into consideration when embarking on such a demanding oncologic scenario.

## 1. Introduction

In 1932, Henry K. Pancoast—a radiologist at the Hospital of the University of Pennsylvania (USA)—first reported the radiological and clinical characteristics of pulmonary tumors arising in the apex of the lung and infiltrating closer structures of the thoracic outlet [1]. In 1961, Shaw and Paulson first described encouraging results by treating this type of tumor with preoperative radiotherapy, followed by pulmonary resection via a wide posterior thoracotomy extending to the neck, which is subsequently called the “Paulson” approach [2]. This approach then remained the only valuable therapeutic option for many years, given the wide surgical field it provided and the possibility to perform a radical resection of posteriorly located apical tumors of the lung. On the other hand, the “Paulson” approach did not allow completely safe surgical control in the case of anterior located tumors, in particular with subclavian vessel involvement.

In 1979, Masaoka et al. described an anterior approach consisting of a proximal median sternotomy extending anteriorly into the fourth intercostal space and the lower part of the neck, thus providing a wide surgical view anteriorly, but with limited control of posterior structures like the transverse process of the vertebrae, the brachial plexus, and the posterior aspect of the ribs [3]. To overcome these limits, the authors subsequently reported a modification of this technique by adding an extended transaxillary periscapular incision from the seventh cervical vertebra to the mid clavicular line, i.e., “Hook approach”; however, due to the wide extension of the incision and related wound-healing problems, this technique was not popularized, and it is only occasionally adopted nowadays [4]. In our clinical practice, we do not use the “Hook approach”.

In 1993, after a previous experience in 1981 on the basis of the pioneering experience of Cormier regarding the surgical approach to the subclavian artery, Dartevelle et al. described the transclavicular approach, subsequently defined as the “Dartevelle” approach; this technique consisted of an L-shaped anterior cervical incision, followed by the resection of the internal half of the clavicle, providing excellent subclavian vessel exposure, conditioned by not-negligible cosmetic and functional side effects on shoulder mobility [5,6,7].

In 1997, Grunenwald and Spaggiari described the transmanubrial approach, consisting of an L-shaped transection of the manubrium of the sternum with first cartilage resection, allowing for the safe retraction of an osteo-muscular flap, including but preserving the clavicle together with its muscular insertion. As a result, this technique not only provides excellent exposure of the subclavian vessels, but also avoids both cosmetic and functional damage related to clavicle resection [8].

With the advent of minimally invasive techniques, hybrid procedures combining robotic- and video-assisted approaches with chest wall resection have recently been reported [9,10,11,12,13].

## 2. Surgical Anatomy

The thoracic inlet represents the apical aperture of the chest and the boundary between the neck and the mediastinum. The manubrium of the sternum and the first rib cartilage anteriorly, the first thoracic vertebra posteriorly, and the first rib laterally represent the margins of the thoracic inlet. The esophagus, the trachea, and the great vessels providing blood supply to the head and the upper extremities exit the chest through the thoracic inlet. This anatomical district can be partitioned into three regions by considering the posterior, middle and anterior insertions of the scalene muscles on the ribs (first and second ribs) as section planes. The anterior region is defined as “prescalene” and is located anteriorly to the insertion of the anterior scalene muscle onto the first rib; the subclavian vein is located in this space, lying closely behind the clavicle. Moreover, the thoracic duct flows into the jugulo-subclavian confluence, and notably, the phrenic nerve lies on the anterior aspect of the anterior scalene muscle. The middle region is defined as “interscalene”, extending from the posterior border of the anterior to the posterior margin of the middle scalene muscles. This compartment is crossed on the right side by the innominate artery, which provides the right subclavian artery and the right carotid artery; on the left side are the left carotid artery and the left subclavian artery. In addition, in the interscalene space, we find two neurological structures: the trunks of the brachial plexus and the vagus nerve. The posterior region is defined as “extrascalene”, extending posteriorly to the middle scalene muscle: here, we find the posterior scapular arteries and many neurological structures, which are the neural foramina, the stellate ganglion, the accessory spinal nerve, the sympathetic chain, and finally, the vertebral bodies.

## 3. Pathophysiology

As superior sulcus tumors compress or infiltrate the surrounding anatomical structures (which occurs more often), they generate several signs and symptoms, which are cumulatively defined as Pancoast syndrome. Very frequently the brachial plexus infiltration causes arm and shoulder pain; in fact, this symptom is observed in almost all affected patients.

Thoracic pain can also be due to parietal pleura, vertebral bodies, or rib infiltration. Usually, respiratory symptoms are observed at a later stage and, for this reason, the first reported symptoms are often misdiagnosed as orthopedic diseases, thus often leading to a delayed oncologic diagnosis. When C8 T1 roots are widely infiltrated, paresthesia and pain in the fourth and fifth finger is reported, as well as pain in the arm, forearm, and hand. In more advanced cases, the involvement of intrinsic hand muscles might result in fine motor skill impairment and reduced handgrip. In the case of cervical ganglion and sympathetic trunk involvement, hyperidrosis and facial flushing—only of the same side—can be observed because of neurogenic irritation. In more advanced cases, Claude Bernard Horner syndrome (miosis, enophthalmos, and ptosis) is reported, sometimes associated with contralateral hyperidrosis and facial flushing due to hyperactive sympathetic response of the other side (Harlequin syndrome).

## 4. Imaging and Functional Assessment

Preoperative assessment of patients suffering from a superior sulcus tumor consists of medical history, physical evaluation, standard blood tests, cardiological evaluation, spirometry, and—in case of planned pneumonectomy—perfusion lung scanning and cardiopulmonary exercise testing. Proper staging protocols require whole-body computed tomography and positron emission tomography; flexible bronchoscopy is required to obtain histologic diagnosis and to explore the anatomy of the airways. Magnetic resonance imaging and angiographic studies are not routinely required and can be considered in some cases. M1 status, as well as pN3 disease, are oncologic absolute contraindications to lung resection. Clinical N2 disease should be always confirmed by endobronchial ultrasound (EBUS) transbronchial needle aspiration (TBNA), or mediastinoscopy if EBUS TBNA is not available. Patients presenting ipsilateral supraclavicular lymph node metastasis—without proven N2 disease—can be still considered for surgical resection. In fact, it has been shown that patients affected by superior sulcus tumor—with single ipsilateral supraclavicular disease—present a long-term prognosis similar to pN1 patients and much better than pN2 patients. Cardiopulmonary preoperative assessment plays a pivotal role in the whole preoperative program in order to properly stratify surgical risk and estimate expected quality of life after surgical excision. It requires a careful medical history, physical evaluation, standard blood tests, cardiological evaluation, spirometry, and in case of planned pneumonectomy, perfusion lung scanning and cardiopulmonary exercise testing. Predicted preoperative FEV 1 greater than 40% allows for low-risk resection; predicted preoperative FEV 1 ranging from 30% to 40% requires further tests (e.g., cardiopulmonary exercise testing), and functional operability should be determined on a case-by-case basis. Before embarking on such a demanding therapeutic protocol, smoking cessation is considered mandatory, and preoperative respiratory exercises are suggested to reduce the risk of postoperative complications.

## 5. The Anterior Transclavicular Approach (Dartevelle)

This approach was first described by Dartevelle et al. in 1993, reporting 29 patients undergoing complete, en bloc resection of the tumor invading the thoracic inlet, chest wall (first and second ribs), and lung, either through the anterior transcervical approach alone in 9 cases or by adding a posterior thoracotomy in 20 cases. The authors performed 14 wedge resections, 14 lobectomies, and 1 pneumonectomy without any perioperative deaths. They observed encouraging overall 2- and 5-year survival rates of 50% and 31%, respectively, with a median follow-up time of 2.5 years.

In this approach, the patient lies in the supine position; the neck is hyperextended, and the head is turned to the opposite uninvolved side. The skin incision consists of an L-shaped cervicotomy, whose branches follow the anterior border of the sternocleidomastoid muscle (vertical branch) and the inferior border of the internal half of the clavicle (horizontal branch).

The attachment of the sternocleidomastoid muscle on the sternum is divided, thus yielding an excellent view of the superior aspect of the subclavian vein and artery; after a careful inspection, showing that the tumor is deemed resectable, the medial half of the clavicle is resected. The dissection starts with the jugular and subclavian veins and—when operating on the left side—thoracic duct ligation could be required. In the case of innominate vein infiltration for the tumor, the vessel can be resected without the need for vascular reconstruction. The dissection continues with the subclavian artery, often requiring dissection and isolation of the internal mammary artery, the ascending cervical artery, and more rarely, the vertebral artery. If the tumor clearly infiltrates the subclavian artery and subadventitial dissection does not allow radical resection, the vessel should be resected and then reconstructed with a vascular prosthesis connected by an end-to-end anastomosis. In the case of brachial plexus infiltration, which is usually present in true Pancoast tumors, the lower trunk and C8 and T1 nerve roots can be dissected up to the spinal foramen. Subsequently, vertebral bodies, sympathetic chain, and the stellate ganglion are easily exposed and freed from the tumor. Finally, resection of the first two ribs is performed, thus allowing en bloc resection of the neoplasm within the lung parenchyma. In the case of larger tumors widely extending to the posterior aspect of the thoracic inlet, an additional posterior thoracotomy may be required to better resect the chest wall and control the pulmonary vessels [5,6].

## 6. The Anterior Transmanubrial Approach (Grunenwald Spaggiari)

In 1997, Grunenwald and Spaggiari significantly overcame the limits of the transclavicular approach due to the removal of the internal half of the clavicle and section of its muscular insertions, leading to shoulder instability with functional and cosmetic consequences. They described a transmanubrial approach through an L-shaped incision of the manubrium and first costal cartilage resection, allowing wide mobilization of an osteo-muscular flap, including but preserving the entire clavicle and all its muscular insertions. The retraction of the osteo-muscular flap allows for great exposure of the whole subclavicular district, with safe control and resection of vascular and neurological structures involved in anterior apical chest tumors. They reported an initial series of six patients operated on using the transmanubrial approach, in which no wound or bony alterations were reported; all patients received early postoperative shoulder mobilization and did not present any functional problems. On the contrary, in the authors’ previous experience on 20 consecutive patients operated on using the transclavicular approach, 14 patients suffered from postural and esthetic defects, which were severe in 8 patients. The transmanubrial approach consists of an L-shaped cervico-manubriotomy with the section of the first rib cartilage. The sternocleidomastoid muscle is then dissected, and the internal jugular vein is easily exposed. The fibers of the pectoralis major muscle are spared, thus allowing for the isolation of the first costal cartilage and the mammary vessels, which are subsequently ligated and transected. The resulting sternoclavicular flap is then progressively mobilized, allowing for the safe dissection of the posterior half of the clavicle, leaving part of the subclavian muscle on the subclavian vessels, representing an ideal plane of dissection. A lace around the manubrial edge can be used to further elevate the flap. The subclavian vein is widely dissected and mobilized, starting from the jugulo-subclavian confluent (Pirogov); the anterior scalene muscle is transected, thus exposing the underlying subclavian artery with its branches. After these steps, all thoracic apical structures can be controlled and resected as required.

In conclusion, this approach provides wide exposure of apical chest structures, allowing a safer resection of elements involved in apical cancers when needed; the whole clavicle is left in site and no major muscular resection is needed, thus preserving full shoulder girdle function, preventing shoulder instability with related esthetic and functional side effects [8] (Figure 1).

## 7. The Anterior Trap-Door Approach (Masaoka Nomori)

In 1979, Masaoka et al. described a new approach for the resection of apical tumors of the chest. It consists of a proximal median sternotomy continuing anteriorly into the fourth intercostal space and extending to the base of the neck on the same side (Figure 2). This incision provides wide exposure of the tumor, subclavian vessels, brachial plexus, vertebral bodies, and posterior aspects of the ribs. After the retraction of the anterior chest wall (trap door), the subclavian vessels and the fibers of the brachial plexus are isolated; both the vagal and phrenic nerves need to be carefully identified, and if there are no signs of infiltration, they can be retracted and preserved. The eighth cervical and first thoracic nerve roots, which contribute to the ulnar nerve, are dissected, as well as the subclavian vessels, which can be reconstructed (the artery) in the case of tumor infiltration. Subsequently, the scalene muscles are dissected and the infiltrated ribs, which are usually the first, the second, and more rarely, the third ribs, are resected. Some parts of the paravertebral muscles, including the iliocostal, latissimus, and semispinalis cervicalis, may be resected in some advanced cases. After these steps, lung resection can be safely performed with radical lymphadenectomy. Although this approach provides excellent exposure of the upper part of the chest cavity, the subclavian vessels can be safely managed, and lung resection is easily performed, some disadvantages are represented by difficult exposure in the case of deep posterior chest wall infiltration and potential wound healing to the extent of the incision.

In their first report, Masaoka et al. reported their experience only in two cases; in 2014, Nomori et al. reported their 25 years’ experience in 33 patients treated using modified trap-door thoracotomy. The modification from the original Masaoka approach involves the transection of the first rib. After sectioning the apical site of the mammary artery, the first rib is isolated and transected from the chest. This procedure, according to the authors’ point of view, can more easily prevent iatrogenic injury to the subclavian vein. The authors reported only one case of postoperative chylothorax without any residual deformity of the chest wall or clavicle [3,14].

## 8. The Posterior Approach (Shaw Paulson)

This approach requires posterolateral decubitus to expose the scapular region and the posterior aspect of the neck. The first step is a standard thoracotomy to explore the pleural cavity and to confirm resectability by assessing the involvement of the chest wall, lung parenchyma, thoracic inlet, and mediastinal structures. Once the procedure is considered feasible, the incision is continued, running posteriorly under the tip of the scapula, and then vertically between the medial border of the scapula and the vertebral bodies up to the seventh cervical vertebra. The latissimus dorsi and the trapezius muscles are divided first, and the dissection then proceeds by dividing the serratus anterior, rhomboids, and elevator of the scapula muscles. In this phase, both the dorsal scapular nerve and the scapular artery—close to the medial margin of the scapula—should be carefully preserved. When the chest wall is widely exposed, chest wall resection is accomplished first to allow easier subsequent lung resection; en bloc resection of bony structures together with the lung is the best option to avoid incomplete resection, which could result from extrapleural dissection without rib resection. In the vast majority of Pancoast tumors, the first two or three ribs are involved and need to be resected, although a wider resection could be necessary to achieve radicality.

The first rib resection is technically challenging: the middle and anterior scalenus muscle insertions onto the first and second rib need to be divided to expose neurovascular structures—the subclavian vein and artery and the brachial plexus—crossing under the clavicle and above the first rib. To allow a radical and safe resection, the head of each involved rib has to be disarticulated from the transverse process of the corresponding vertebra by dividing the costotransverse ligament. Finally, the corresponding intercostal nerve, which originates from the relative intervertebral foramen, needs to be correctly identified and divided. Care should be taken while coagulating close to the spinal cord to prevent neurological injuries; moreover, hemostatic sponge migration into the spinal cord must be carefully prevented to avoid medullary compression with severe neurological sequelae. The lower trunk of the brachial plexus (C8 T1 nerve roots) is visualized and dissected; most commonly only T1 infiltration is observed, but when C8 is also involved, it is necessary to medially divide the lower trunk of the brachial plexus at its origin from the spine. Subsequently, the first rib is resected, thus making it possible to release the chest wall from the apex of the thorax: this allows for division of the lower part of the stellate ganglion, as well as the first intercostal artery. In the vast majority of cases, chest wall reconstruction is not required because of the protection of the scapula; when a more extended chest wall resection has been performed, i.e., removing more than the first three ribs, chest wall reconstruction via methylmethacrylate rigid prosthesis is recommended to prevent a trapped scapula [15,16,17,18,19]. Vertebral body infiltration is usually considered a contraindication for surgical resection of Pancoast tumors; however, some experiences have been reported, combining thoracic and orthopedic experience [20,21] (Figure 3).

## 9. Hemiclamshell Approach

This is a combination of anterior thoracotomy in the fourth intercostal space (on the left or right side) with longitudinal median sternotomy extended from the manubrium to the fourth sternocostal joint. It requires mammary vessel ligation on the side of the anterior thoracotomy. The posture of the patient varies according to the planned resection: in the case of mediastinal involvement without any major airway infiltration, the supine position with 90° bilateral arms abduction is indicated. On the contrary, when major pulmonary resection is expected, the lateral or semilateral position should be preferred.

Anterolateral thoracotomy is performed first in order to assess neoplasm extension, as well as structure infiltration, to confirm resectabilty; if the procedure is judged feasible, mammary vessel ligation is performed, and median sternotomy is then easily completed. This allows for the creation of a sternocostal flap, which is then gradually retracted to obtain excellent exposure both of the mediastinum and the involved pleural cavity. There is no need to approach the sternoclavicular joint, which is left untouched. This approach makes it possible to easily control the take-off from the aortic arch of the subclavian and carotid arteries, as well as the jugulo-subclavian vein confluence and the innominate vein, on the left side; on the right side, take-off from the jugulo-subclavian vein confluence and the superior vena cava can be controlled.

On the contrary, this incision does not allow for ideal control of the distal part of the supra-aortic branches and venous vessels, as well as of the posterior aspect of the chest wall; the hemiclamshell approach should thus be reserved for apical lung or mediastinal tumors, which only proximally infiltrate the venous and arterial branches [14,18,19,20].

## 10. Hybrid Approaches

With the advent of minimally invasive thoracic surgical approaches, hybrid techniques combining chest wall resection with video-assisted or robotic-assisted lung resection have been recently proposed [9,21]. Uchida et al. reported a case of right upper lobectomy en bloc with chest wall resection for a lung squamous cell carcinoma infiltrating from the second to the fourth rib; pulmonary resection was accomplished using the da Vinci Xi system together with a 15 cm posterior thoracotomy. The erector spinae muscles were fully preserved, while the second to fourth ribs were posteriorly transected at the level of the costovertebral joint. Subsequently, the second to fourth ribs were divided from the lower aspect, and the upper border of the second rib was crossed forward, accomplishing the chest wall resection. No intraoperative complications were reported; the postoperative course was uneventful, and the patient was successfully discharged on postoperative day eight [9]. Mariolo et al. reported a case of squamous cell carcinoma of the left upper lobe invading the thoracic outlet, treated with a left upper lobectomy with chest wall resection, combining an anterior transmanubrial approach with a left upper lobectomy, performed using the da Vinci Xi system (Table 1).

## 11. Oncologic Considerations

The morbidity and mortality rates after Pancoast tumor treatments range from 10 to 55% and 0 to 7%, respectively [22]. The 5-year survival rate has significantly improved in recent years thanks to the advancement of treatment options [23]. A few papers investigated the prognostic factors of Pancoast tumors, including surgical margin radicality, pathologic response to induction treatments, tumor stage (T status), and lymph node involvement (N status) [24]. Radical resection of true Pancoast tumors is difficult to achieve because of anatomical and technical difficulties, although it has been demonstrated that induction chemoradiotherapy increases the radical resection rate [24]. An intergroup prospective phase 2 trial (SWOG-9416/INT-0160) tried to maximize the favorable effects of medical therapy by adding cycles of etoposide and cisplatin, but patients were not able to complete the adjuvant treatment proposed [25]. The Southwest Oncology Group (SWOG) S0220 trial introduced two additional cycles of docetaxel, a consolidation chemotherapy: this led to better local control rates without any impact on distant metastases, in particular brain metastases, which significantly affected overall survival [26].

In this study, 46 patients with histologically proven Pancoast tumors were enrolled; they were staged as T3 or T4 non-small-cell lung cancers, N0 to N1 without any distant metastases (M0). They received induction treatment with cisplatin and etoposide, plus concurrent 45 Gy radiation therapy. Patients who did not show any progression received surgical treatment within seven weeks. Consolidation treatment was delivered with docetaxel every three weeks; the primary endpoint of the study was feasibility. Of the 46 registered patients, 44 were finally enrolled; 86% completed standard induction treatment; 66% received lung resection; and only 45% completed consolidation therapy with docetaxel. A total of 97% of operated patients (28 out of 29) received radical resection (R0), and among them, 2 patients (7%) died because of acute respiratory distress syndrome. The authors observed a complete or almost complete pathologic response in 21 of 29 operated patients (72%). Local recurrence was observed in two patients, systemic recurrence was reported in ten patients, and local and systemic recurrence was observed in one patient. Among the ten patients who presented systemic relapse, seven disclosed brain-only recurrence [26].

## 12. Conclusions

Pancoast tumors still represent a complex clinical condition requiring considerable technical surgical skills within more articulated multimodality treatment.

Although a multimodality approach combining chemotherapy, radiotherapy, and surgery allows for radical resection and an effective local control in the vast majority of patients, many patients cannot receive surgical resection or complete the whole programmed therapeutic regimen. Systemic relapse, particularly cerebral recurrence, still poses a significant issue in this cohort of patients. Global clinical condition, tumor histology, and long-term perspectives should always be taken into consideration when embarking on such a demanding oncologic scenario [27,28,29,30,31,32,33,34,35,36,37,38].

## Figures and Tables

**Figure 1 jpm-13-01168-f001:**
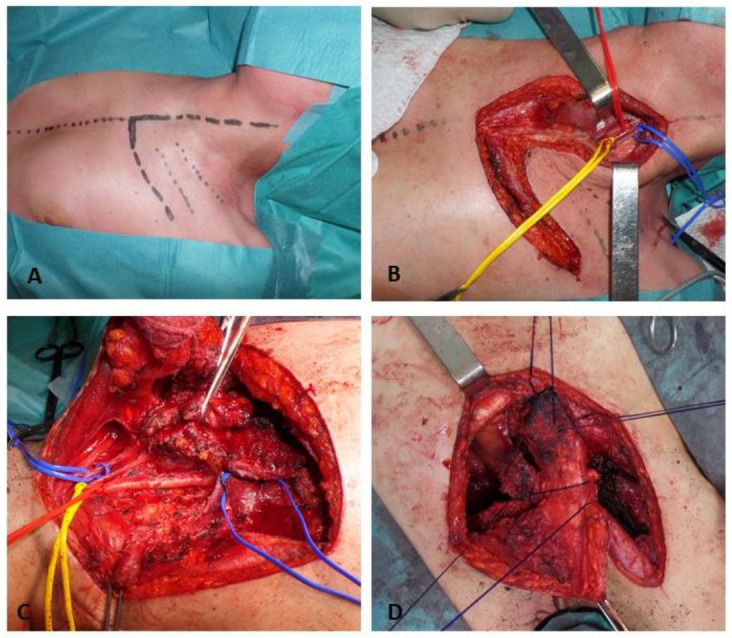
(**A**) Patient decubitus; (**B**) Cervicothoracic incision; (**C**) Vascular dissection; (**D**) Costoclavicular joint reconstruction.

**Figure 2 jpm-13-01168-f002:**
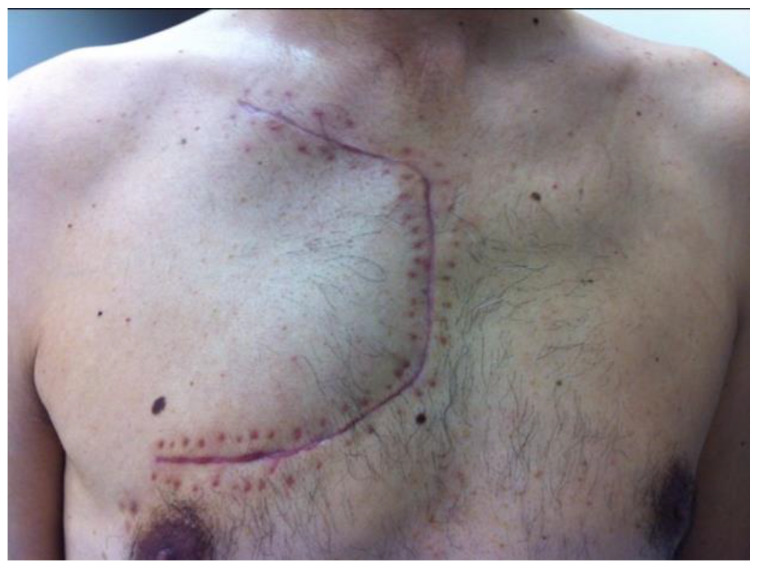
The Masaoka Nomori approach (trap door).

**Figure 3 jpm-13-01168-f003:**
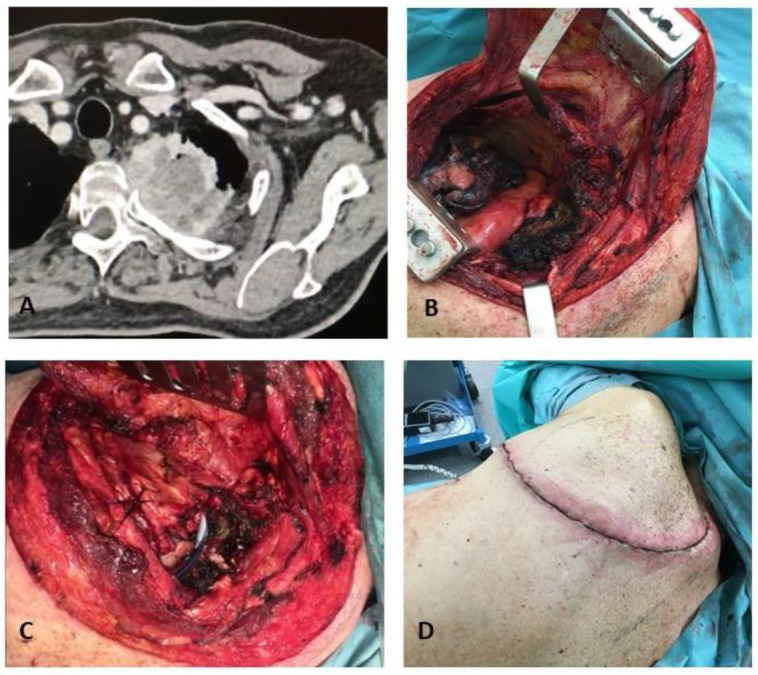
(**A**) Costovertebral joint infiltration at preoperative CT scan; (**B**) surgical view after costovertebral disarticulation; (**C**) chest wall closure; (**D**) postoperative view.

**Table 1 jpm-13-01168-t001:** Pros and contras of surgical approaches to Pancoast tumors.

Approach	Strength Points	Weak Points
Transclavicular (Dartevelle)	Exposure of subclavian vessels	Difficult exposure of vertebral bodies Shoulder functional impairment
Transmanubrial (Grunenwald Spaggiari)	Exposure of subclavian vessels	Difficult exposure of vertebral bodies
Trap door (Masaoka Nomori)	Exposition of subclavian vessels, the whole mediastinum and pleural cavity	Difficult exposure of vertebral bodies Healing problems due to multiple incisions
Posterior (Paulson)	Exposition of vertebral bodies and brachial plexus	Difficult exposition of subclavian vessels
Hemiclamshell	Exposition of subclavian vessels, the whole mediastinum and pleural cavity	Difficult exposure of vertebral bodies Healing problems due to multiple incisions

## Data Availability

The data can be shared up on request.

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
