# Peer review of "Surgical Approaches to Pancoast Tumors"

_jpm, 2023, doi:10.3390/jpm13071168_

Round 1

Reviewer 1 Report

Your submitted manuscript entitled "Surgical Approaches to Pancoast Tumors" is a valuable narrative review describing, in a concise manner, all methods of surgical resection of Pancoast tumours described to date.

I find this manuscript, written by eminent experts and authors of one of the described anterior approaches, very interesting and relevant to JPM readers, especially to fellow thoracic surgeons preparing for their final exams.

I recommend publication of the manuscript in its present form.

Author Response

Response to Reviewers Reviewer 1 Your submitted manuscript entitled "Surgical Approaches to Pancoast Tumors" is a valuable narrative review describing, in a concise manner, all methods of surgical resection of Pancoast tumours described to date. I find this manuscript, written by eminent experts and authors of one of the described anterior approaches, very interesting and relevant to JPM readers, especially to fellow thoracic surgeons preparing for their final exams. I recommend publication of the manuscript in its present form. Thank you for reviewing our paper.

Reviewer 2 Report

Overall well structured article, however some shortcomings are there which  mentioned below: 

1. In whole manuscript at many places authors put "-" (dash) which is not appropriate. 

2. In abstract, line no. 10-12, author may rephrase this sentence. 

3. In introduction, line no 33, Author may rephrase this statement as this sentence started with digit, which is not appropriate.  

4. At many places "et coll" written, which should be "et al". 

5. In page no.2 line no. 48. Author discussed about "Hook Approach". Author may provide some information about the specific conditions when Hook approach considered for the treatment option. 

6. There is no information about the source of images. Author should provide appropriate citation of the images.

7. Author may put a table for a all available treatments for Pancoast Tumors, what are their lacuna, when it came into the practice and which one the most acceptable and advanced methodology. How many countries ( list of country) have these type of facility.

There are lots of - (dash) and other typography error in the manuscript. At few places need to rephrase the statements to furnish a sound statement. 

Author Response

Reviewer 2 Overall well structured article, however some shortcomings are there which mentioned below: 1. In whole manuscript at many places authors put "-" (dash) which is not appropriate. We removed “-“ (dash) where it was inappropriate 2. In abstract, line no. 10-12, author may rephrase this sentence. We rephrased the sentence in this way: “Surgical resection still plays a pivotal role within the multimodality approach.” 3. In introduction, line no 33, Author may rephrase this statement as this sentence started with digit, which is not appropriate. We rephrased the sentence in this way “In In 1961, Shaw and Paulson first described…” 4. At many places "et coll" written, which should be "et al". We changed “et coll” into “et al” 5. In page no.2 line no. 48. Author discussed about "Hook Approach". Author may provide some information about the specific conditions when Hook approach considered for the treatment option. In our clinical practice we do not use the “Hook approach”. We reported this sentence “however, due to the wide extension of the incision and related wound healing problems, this technique was not popularized and it is only occasionally adopted nowadays [4]. In our clinical practice we do not use the “Hook approach”. 5. There is no information about the source of images. Author should provide appropriate citation of the images. We reported in the Acknowledgments section Acknowledgments: All the intraoperative views come from surgical operations performed by prof. Francesco Petrella at the European Institute of Oncology. 6. Author may put a table for a all available treatments for Pancoast Tumors, what are their lacuna, when it came into the practice and which one the most acceptable and advanced methodology. How many countries (list of country) have these type of facility. We added a new table one focusing of pro and contras of each surgical approach We removed ref 34 and 35 beacuse too old, as requested by the editor